# Inhibitory Effect against *Listeria monocytogenes* of Carbon Nanoparticles Loaded with Copper as Precursors of Food Active Packaging

**DOI:** 10.3390/foods11192941

**Published:** 2022-09-20

**Authors:** Adriana Scattareggia Marchese, Elena Destro, Carlo Boselli, Francesco Barbero, Mery Malandrino, Giusy Cardeti, Ivana Fenoglio, Luigi Lanni

**Affiliations:** 1Istituto Zooprofilattico Sperimentale del Lazio e della Toscana “M. Aleandri”, Sede di Roma, Via Appia Nuova 1411, 00178 Rome, Italy; 2Department of Chemistry and Interdepartmental Centre for Nanostructured Interfaces and Surfaces (NIS), University of Turin, 10125 Turin, Italy

**Keywords:** *Listeria monocytogenes*, shelf-life of food, carbon nanoparticles, minimum inhibitory concentration (MIC), antimicrobial activity, copper-loaded nanoparticles, active food packaging, action mechanism

## Abstract

Human listeriosis is a serious foodborne disease of which outbreaks are occurring increasingly frequently in Europe. Around the world, different legal requirements exist to guarantee food safety. Nanomaterials are increasingly used in the food industry as inhibitors of pathogens, and carbon nanomaterials are among the most promising. In the present study, novel carbon nanoparticles loaded with copper (CNP-Cu) were prepared, and their antimicrobial activity against *Listeria monocytogenes* was assessed. CNPs of two sizes were synthesized and characterized by dynamic light scattering (DLS), electrophoretic light scattering (ELS) and electron microscopy (EM). The minimum inhibitory concentration (MIC) of CNP-Cu was determined in accordance with the available standard. To get insights into its mechanism of action, the release of copper ions into a cell media was assessed by inductively coupled plasma optical emission (ICP-OE), and the ability of loaded CNPs to generate cytotoxic reactive oxygen species (ROS) was evaluated by EPR spectroscopy. Finally, the extent of release of copper in a food simulant was assessed. The results demonstrated the antimicrobial effectiveness of CNP-Cu, with growth inhibition up to 85% and a release of copper that was more pronounced in an acidic food simulant. Overall, the results indicate CNP-Cu as a promising agent for the design of active food packaging which is able to improve food shelf-life.

## 1. Introduction

The safety of food products is a major concern for the food industry. Pathogen microorganisms, whose metabolites have been identified as significant hazards for human health, represent a risk in cases of contaminated food. Among them, the FDA considers *Escherichia coli, Salmonella* spp., *Campylobacter* spp., *Staphylococcus aureus* and *Listeria monocytogenes* (*L. monocytogenes)* as the most problematic [1]. In particular, *L. monocytogenes* is one of the main threats to the safety of food products due to its ability to persist in establishment environments (non-food contact surfaces and food contact surfaces) [2]. The pathogen causes a disease called listeriosis; outbreaks of this disease are becoming increasingly frequent Europe. The average lethality rate is estimated to be around 15–16%, both in the European Union and in the USA. Moreover, for immunocompromised individuals, the lethality rate can reach 30–40% [3]. This has prompted the development of new strategies to limit pathogen growth and spread, as well as the use of antibiotics in food [4]. Food contact materials (FCM) and food packaging science have made great advances in recent decades, shifting their focus from maintaining to enhancing food quality, according to various food safety criteria. The development of active and smart materials has led to a new way of dealing with modern needs. These are not inert systems, but rather, make use of the properties of their design and components to actively monitor, improve and extend the shelf-life of food products. Among them, nanomaterials have emerged as novel antimicrobial agents. Applications of such technology reported in literature have mainly focused on antimicrobial nanoparticles (NPs), such as Ag, Ag_2_O, ZnO, Cu, Au, TiO_2_, lipidic and carbon-based NPs [5,6,7,8,9,10].

Carbon-based nanomaterials are among the most widely studied and used due to their exceptional properties like thermal stability and conductivity, as well as their possible surface functionalization with other materials [11]. Several systems based on carbon nanomaterials have already been developed with antimicrobial goals, proving their bactericidal or bacteriostatic effect against many species like *E. coli, S. aureus, E. faecalis, A. niger* and *Salmonella* spp. [12,13,14,15,16,17,18,19]. On the other hand, their availability in the food market is still very limited, because their ingestion presents a potential risk for human health which still needs to be investigated.

The antimicrobial properties of copper have been known since ancient times [20]. Today, there is a large body of evidence supporting its efficacy against different species of G+ and G− microorganisms, including *Listeria monocytogenes* [21,22,23,24,25,26]. On the other hand, the mechanism of action of Cu is still under discussion. In this regard, its abilities to generate oxidative stress, to inactivate key cellular pathways, to induce protein misfolding and aggregation and to induce plasma membrane damage have been proposed [20,27,28,29]. The most important initial event in aerobic conditions is thought to be the generation of reactive oxygen species (ROS) by Cu(II)/Cu(I) redox cycling [20,27,30].

Copper exhibits antimicrobial properties not only as a soluble salt (e.g. CuSO_4_), but also in elemental form [21], or in the form of Cu or CuO nanoparticles [31,32,33]. In the latter form, its use has been proposed as an active agent in composites for food packaging [34,35]. For this application, safety is an important issue, and the release of copper by food packaging is regulated [36].

Some studies have reported the preparation and antimicrobial activity of nanocomposites based on carbon nanomaterials like reduced graphene [29] or carbon nanotubes [37] with Cu or CuO nanoparticles. To the best of our knowledge, to date, no study has assessed the antimicrobial activity against *Listeria monocytogenes* of copper associated with carbon nanomaterials.

The present study investigates the antimicrobial efficacy of novel nanostructures composed of glucose-derived carbon nanoparticles—a green, sustainable, low-cost type of carbon nanomaterials [38]—loaded with copper (CNP-Cu) against a *Listeria monocytogenes* reference strain (ATCC 19117). As the available pharmaceutical minimum inhibitory concentration (MIC) [39,40] and enumeration protocols for *L. monocytogenes* were not suitable for our experimental conditions, a new MIC protocol that combines inhibition in solution and the ISO horizontal method [41] was developed. To gain insights into the mechanism of action of this CNP-Cu system, similar concentrations of soluble copper were also tested, while the ability of CNP-Cu and soluble copper to generate ROS was evaluated by EPR spectroscopy. The release of copper in food simulants was also investigated to assess the safety of these systems as potential active components of food packaging materials.

## 2. Materials and Methods

### 2.1. Reagents

D-(+)-glucose, sodium polyacrylate, acetic acid 99–100% (glacial), nitric acid (≥65%), hydrogen peroxide 30% (*w*/*w*), Triton X-100, sodium tartrate dehydrate, sodium hydrogen carbonate, ascorbic acid and phosphate buffered saline powder were obtained from Sigma-Aldrich (Steinheim, Germany). Copper (II) sulfate-pentahydrate and ethanol were purchased from Merck (Darmstadt, Germany); bicinchoninc acid disodium salt and sodium carbonate decahydrate from Alfa Aesar (Haverhill, MA, USA); and 5,5-dimethyl-1-pyrroline-N-oxide (DMPO) from Enzo Life Science (Farmingdale, NY, USA). The olive oil (a mixture of refined and virgin olive oils) was purchased in a supermarket in Italy. Ultrapure water was obtained from a Milli-Q Plus system (Millipore, Bedford, MA, USA) and was always used freshly prepared.

Brain heart infusion broth (BHI), listeria selective agarb (LSM), and blood agar plates were produced by Biolife, while plate count agar (PCA) was produced by Biokar diagnostics. The reference strain for experimental trials was the American Type Culture Collection (ATCC 19117), which came from a *Listeria monocytogenes* (serotype 4d) third line culture to prevent the risk of phenotypic alteration.

### 2.2. Synthesis, Loading with Copper and Purification of CNPs

CNP synthesis was carried out in a simple, one-step process based on the hydrothermal carbonization method proposed by Kokalari et al. [42]. The synthetic parameters were modified to obtain CNPs of two different dimensions, referred herein as small (**S**CNP) and large (LCNP). For the SCNP, 2.0 g of glucose was dissolved in 50 mL of ultrapure water, and 15 mg of sodium polyacrylate was added to prevent the crosslinking of the nanoparticles during the synthesis. The solution was magnetically stirred for 45 min at 60 °C in an oil bath to better mix the reagents. It was then stored overnight at 4 °C. Next, it was transferred into a Teflon-lined stainless-steel autoclave (Büchi AG, Uster, Switzerland, volume 100 mL, −1/+100 bar, −10/+300 °C ) and placed in a preheated oven at 190 °C for 3 h. After the synthesis, the CNP suspension was purified with ultrapure water by ultrafiltration, using Vivaflow® 50R cassettes (Sartorius, Stonehouse, UK, 30 kDa cutoff).

For the synthesis of the LCNP, 5.0 g of glucose was dissolved in 50 mL of ultrapure water Milli-Q following the above procedure; the reaction continued for 8 h at 190 °C. At the end, the LCNP suspension was centrifuged at 15,000× *g*, 20 °C, for 30 min (by ROTINA 380R, Hettich, Westphalia, Germany), and the pellet was resuspended in the same volume of Milli-Q water. The operation was repeated three times to purify the synthesis product. The CNP concentration was evaluated by a gravimetric procedure. A volume of 1 mL of the suspension was added into glass vials which had been weighed before filling. The vials were kept in an oven at 70 °C overnight to dry the suspension and were then weighed after evaporation to calculate the final concentration of the nanoparticles. This procedure was performed in triplicate for each suspension. Concentration values were expressed as mg of CNP/mL of suspension. The yield was higher for LCNP (33.46 mg/mL) than for SCNP (1.53 mg/mL), i.e., to 33% and 4%, respectively, relative to the precursor. Both the SCNP and LCNP samples underwent a copper ion loading process incubating 50.3 mL of CNP suspension with 4.7 mL CuSO_4_ aqueous solution (3.10 × 10^−1^ M) and were stirred for 30 min at room temperature [43]. The ratio between CNP and CuSO_4_ solution was in line with that used by Tacu et al. in a previous study [11], which achieved satisfactory loading results. To remove the excess metal ions, the suspensions were centrifuged at 15,000× *g*, 25 °C, for 30 min (ROTINA 380R, Hettich, Westphalia, Germany). Then, 95% of the supernatant was discarded and the pellet was resuspended in the same volume of Milli-Q water; this procedure was performed three times. Part of SCNP and LCNP suspensions were conserved in their pristine form as controls and submitted to the same centrifuging/resuspension operations to avoid artefacts.

### 2.3. Physico-Chemical Characterization of CNPs

As the EFSA journal guidelines [44] state, the size of nanomaterials should be measured by at least two independent techniques, with one being electron microscopy. Here, the size distribution of CNPs was assessed using transmission electron microscopy (TEM), field emission scanning electron microscopy and light scattering techniques. The ζ-potential was determined as an index of nanoparticle surface charges.

The hydrodynamic diameter (dH), polydispersity index (PdI) and ζ-potential of both pristine and copper-loaded CNP suspensions were measured in ultrapure water using Zetasizer (Nano ZS Malvern Instruments, Worcestershire, UK) instrument. A volume of 0.1 mL of CNP suspension was diluted at ratios of 1:10 (SCNP) or 1:200 (LCNP) in ultrapure water and sonicated by probe sonication for 2 min, 30% amplitude (Bandelin Sonopuls HD 3100, Berlin, Germany). The resulting suspensions were transferred to a polystyrene cuvette and analyzed by a dynamic light scattering (DLS) technique. The same mixture was then transferred into a disposable folded capillary cell (DTS 1070) and analyzed by an electrophoretic light scattering (ELS) technique. The results were expressed as mean values of five independent measurements; setting measurement angle = 173 °C; temperature = 25 °C; equilibration time = 120 seconds. The parameters used for the carbon materials were as follows: refractive index = 2.420; and absorption = 1.00. The parameters used for the water dispersant were as follows: refractive index = 1.330; viscosity (25 °C) = 0.8872 cP; and dielectric constant = 78.5.

The geometrical diameter and shape of the synthesized particles were determined by field emission scanning electron microscopy (FE-SEM) analysis using a TESCAN S9000G (TESCAN ORSAY HOLDING, Brno, Czechia) equipped with a Schottky type source. Samples were prepared by drop-casting 10 μL of the sample onto a slide pre-metallized with 5 nm chromium and left to dry at room temperature. At least 300 particles from different regions of the slide were measured and counted. TEM analyses were carried out using a Philips EM 208 electron microscope. Images were taken at 18 × magnification at 80 kV. Samples were prepared using 400 mesh copper grids covered with carbon reinforced plastic film. Before use, each grid was subjected to Alcian Blue staining to ensure the hydrophilic behavior of the grids. The sample were diluted at a ratio of 1:100 in distilled water (pH 7.0). A volume of 50 μL of sample was pipetted onto a piece of dental wax and then a grid support was kept in contact with the sample with the copper side facing upward for 15–20 min. Each grid was exposed to a drop of 2% of phosphotungstic acid (pH 6.6) for 2 min to counter-stain it; then, the excess material was removed [45]. CNPs were evaluated for size, shape and available surface morphology at 56 × magnification. Analysis times were standardized: either 15 min viewing time of the sample grid or checking 15 grids squares, whichever was shortest.

### 2.4. Evaluation of the Amount of Copper Loaded onto CNPs

Five milliliters of each CNP suspension was transferred to high-pressure Teflon bombs and 1 mL H_2_O_2_ (30% *w/w*) and 4 mL HNO_3_ (≥65%) were added. Digestion was carried out using a microwave digestion system (Milestone microwave MLS 1200 Mega, Sorisole, Italy) as follows: 1 min 250 W, 2 min stop, 5 min 200 W, 5 min 350 W, 5 min 550 W, 5 min 250 W, 10 min cooling. Microwave digestion was also performed with only the mixture of HNO_3_ and H_2_O_2_ as a blank. The resulting solutions were filtered, transferred into plastic containers and diluted to 15 mL with ultrapure water. The amount of metal ions in each sample was determined by the inductively coupled plasma optical emission spectrometry technique (ICP-OES) (PerkinElmer Inc. Optima 2000 DV, Norwalk, CT, USA).

### 2.5. CNP-Cu Microbiological Sterility Tests

The synthesis of CNPs was not carried out under microbiological laboratory conditions, i.e., according to ISO 17025. Therefore, before testing, microbiological contamination (bacteria yeast and mold) was assessed at regular intervals. Tests were carried out in accordance with Section 8 of ISO 7218 [43].

In order to obtain a benchmark on the microbiological contamination of the suspensions, tests were carried out using both pristine and CNP-Cu. All experimental analyses were performed according to general requirements for microbiological testing laboratories (ISO 17025, ISO 7218) and for the preparation, production, storage and performance testing of culture media (ISO 11133) [43,46,47]. The spreading-spatula method was used to identify any contamination on plate count agar and blood agar as non-selective microorganism culture media. Plates were incubated at 30 ± 2 °C for 5 days. To monitor CNP sterility, tests were repeated at defined sampling points (ISO 20976-1 paragraph 3.19), during which antimicrobial activity tests were performed and CNP-Cu sterility was assessed before each MIC test [48]. Sterility testing was carried out for 143 days before MIC tests. LCNP-Cu were not contaminated during the testing period, but SCNP-Cu showed low-level contamination after 88 days. The SCNP-Cu samples were sterilized and used for the experimental trials. SCNP-Cu and LCNP-Cu were stored and used under the same environmental conditions.

### 2.6. Listeria Monocytogene Growth Trend Analysis

The growth features of *Listeria monocytogenes* depend on their strain type and matrix composition (food or broth). The growth parameters for *Listeria monocytogenes* (MURRAY et al.) Pirie ATCC19117 [49] used in this work were 30 ± 2 °C for resuscitation and 37 ± 2 °C for assessing the efficacy of the antimicrobial materials [2].

More details regarding references and other information relating to the reference material are available on American Type Culture Collection website (www.atcc.org accessed on 22 June 2022). The inoculum was prepared from ATCC 19117 cryobank® beads which were re-hydrated and spread as reported in [50], including two main steps (re-hydratation, growth in BHI broth) to maximize the recovery cells and achieve the reference stock (ISO 11133 paragraph 3.4.3) [47]. To obtain more information on its growth features, exponential and stationary phases were determined over time according to ISO 20976-1 [48]. The growth trend assessment was carried out at specific intervals and the enumeration was carried out in accordance with ISO 11290-2 [41].

### 2.7. Preliminary Tests on a Suitable Volume of Growing Broth

The MIC protocol was performed, maintaining a fixed final volume of MIC solution. Therefore, to test the increasing concentrations of antimicrobial solutions (Table 1), it was necessary to decrease the broth volume in order to maintain a constant CFU/mL of *L*. *monocytogenes*. To ensure that optimal growth conditions were used, *L. monocytogenes* growth was evaluated in decreasing broth volumes (with reduced nutrient conditions), in line with ISO 6887-1 [51]. A suitable volume of broth culture was used as a control, while the growth of *Listeria monocytogenes* was evaluated in 900 μL, 800 μL and 700 μL BHI and by adding peptone water to achieve a final volume of 1000 μL. Microorganisms growth was evaluated by three experimental trials, executed at different time points. According to the following MIC protocol, a concentration of 10^5^ CFU/mL was considered as a starting point, and growth was examined after 24 h of incubation at 37 ± 2 °C.

### 2.8. MIC Protocol

The growth trend analysis of ATCC 19117 allowed us to set up a suitable MIC protocol to satisfy the required needs (Appendix A).

The MIC protocol used was based on the maintenance of fixed concentration of microorganisms in contact with different volumes of antimicrobial solutions in order to identify the minimum volume that was able to inhibit the growth of the microbial population.

The MIC was determined in accordance with the standard proposed in ISO 20776-1 [52], and according to the Clinical Laboratory Standard Institute (CLSI) [39] and the European Committee on Antimicrobial Susceptibility Testing (EUCAST) [40] in order to obtain a protocol which was able to meet the required needs. The inoculum was prepared following the direct colony suspension method (ISO 20776-1) [52] and blood TSA agar medium was used to achieve single colonies after incubation at 37 ± 2 °C for 24 h (ISO 11133 paragraph t 5.4.2.1) [47]. Selected colonies (2 or 3) of *Listeria monocytogenes* reference stock were touched with a sterile loop and incubated overnight in BHI broth at 30 ± 2 °C in order to obtain 10^9^ CFU/mL. The suspension was diluted in peptone water (PTW) through decimal dilution series to achieve 10^5^ CFU/mL, as indicated in ISO 6887-1, and enumerated following the procedure reported in ISO 11290-2 [41,51]. The working culture (ISO 11133 paragraph 3.4.5) concentration of CFU/mL was calculated and confirmed for each trial [47]. The MIC test was executed in BHI for 24 h at 37 ± 2 °C.

In order to test a standard CFU/mL of *L. monocytogenes* and different antimicrobial concentrations (Table 1), we decided to maintain a standard final reaction volume of 1000 µL. The MIC solution was made up of 100 µL of 10^5^ CFU/mL, 100 µL or 200 µL or 300 µL of antimicrobial solution and BHI. To obtain mono-dispersed nanoparticles, each suspension was sonicated before each trial. A Soniprep 150, set at 10 Hz for 2 min in ice, was used for the sonication procedure. After the incubation of the MIC solution, the procedure reported in paragraphs 9.2 and 9.3 of ISO 11290-2 was used to enumerate *L. monocytogenes* in both the negative control and the MIC solution [41].

Both small and large CNP-Cu and CuSO_4_ · 5 H_2_O (Cu^2+^) were tested as antimicrobials. The antimicrobial activity of soluble copper was compared to that of the carbon nanoparticles loaded with copper. To obtain comparable results between soluble copper and copper loaded on nanoparticles surface, the concentration of copper on the surface of the nanoparticles was measured by ICP-OES (Table 2). Then, equivalent, increasing soluble copper (low, medium, high) concentrations were tested on *Listeria monocytogenes* ATCC 19117 (Table 1) for each sample.

### 2.9. Quantification of Spontaneous Copper Migration from CNP-Surface to BHI

Copper migration tests were carried out in brain heart infusion (BHI) broth by adding increasing amounts of CNP suspension, i.e., 1 mL, 2 mL and 3 mL, to the broth, maintaining a final reaction volume of 10 mL. A sample containing BHI broth only was used as a control. All the samples were incubated at 37± 2 °C for 24 h and then centrifuged at 8250× *g* 4 °C for 50 min (by ROTINA 380R, Hettich, Westphalia, Germany). The free-nanoparticle supernatants were collected and stored at −20 °C. The quantification of the copper spontaneously released in the broth was performed following the method described in Paragraph 2.4, i.e., by adding 1 mL H_2_O_2_ and 5 mL HNO_3_ to 0.5 g of each BHI sample containing copper ions and the control. The digestion program used was as follows: 1 min 250 W, 1 min stop, 5 min 250 W, 5 min 400 W, 5 min 650 W and 20 min cooling. A mix of HNO_3_ and H_2_O_2_ was prepared as a blank. The resulting solutions were diluted to 15 mL with ultrapure water and the amount of copper ions in each sample was determined by ICP-OES, as indicated in Paragraph 2.4.

### 2.10. Evaluation of Redox Activity of CNPs by Electronic Paramagnetic Resonance (EPR)

Reactivity tests for SCNP, SCNP-Cu, LCNP and LCNP-Cu were carried out by EPR spectroscopy/spin trapping technique using a MiniScope MS100 spectrometer (Magnettech, GmbH, Berlin, Germany). Experiments with DMPO were performed to monitor the production of hydroxyl radicals via Fenton-like reaction. SCNP-Cu and LCNP-Cu suspension were transferred to a glass vial with 0.1 mL PBS (10 mM, pH 7.4), 0.25 mL DMPO solution (0.176 M, in water) and 0.1 mL H_2_O_2_ solution (0.2 M, in PBS 10 mM pH 7.4). The volume of suspension used was determined to ensure that the concentration of copper in the final reaction volume was 1.15 ppm for both samples. SCNP and LCNP were analyzed as controls by adding volumes containing the same mass of nanoparticles as SCNP-Cu and LCNP-Cu. The same reaction was performed in the absence of nanoparticles (negative control) and in the presence of 1.15 ppm of CuSO_4_ (positive control). Measurements were performed after 10, 30 and 60 min from the beginning of the reaction, triggered by the addition of the H_2_O_2_ solution, by immediately transferring a minimal volume of the reaction mixture into a 50 μL glass capillary and analyzing it. Three independent experiments were carried out for each reaction. To collect information about the oxidation state of the copper inside the reaction mixture, 0.025 mL CuSO_4_ solution 0.36 mM was mixed with the same volume of either water or 3.6 mM ascorbic acid solution in order to completely reduce all the Cu^2+^ ions to Cu^+^. The same DMPO reaction mixture as described above was added to both samples, which were then analyzed after 10, 30 and 60 min.

### 2.11. Evaluation of the Reduction of Copper through Bicinchoninic Acid (BCA) Assay

The oxidation state of copper was also monitored by bicinchoninic acid (BCA) assay. To this end, 0.61 g of bicinchoninic acid, 2.7 g of Na_2_CO_3_ · 10 H_2_O, 0.095 g of sodium tartrate dihydrate and 0.475 g of NaHCO_3_ were dissolved in 50 mL of Milli-Q water, and the pH of the solution was adjusted to 10.3 using 1 M NaOH. Then, 0.5 mL of 0.36 mM CuSO_4_ solution was added to 0.5 mL of BCA mixture and alternatively with either 0.1 mL 3.6 mM ascorbic acid solution or 0.1 mL of H_2_O_2_ solution (0.2 M, in PBS 10 mM pH 7.4). The appearance of a purple color indicated the presence of Cu^+^ ions.

### 2.12. Incubation of SCNP and SCNP-Cu in Food Simulants

The migration tests were carried out in ethanol 10% (*v*/*v*), acetic acid 3% (*w*/*v*) and olive oil, in line with Commission Regulation (EU) No 10/2011 [36]. Migration in ultrapure water was also monitored as a control. The samples were placed in a preheated oven and incubated at 40 °C for 10 days; this is conventionally considered to be the most severe conditions of time and temperature. At the end of the incubation, the SCNP and SCNP-Cu were completely removed from the food simulants and the fluids were stored at −20 °C.

#### 2.12.1. Aqueous Food Simulants

A volume of 5 mL of each CNP suspension was added to a specific standard simulant up to a final volume of 20 mL and left at 40 °C for 10 days. Then, all samples were centrifuged at 6000× *g*, 25 °C for 45 min (by ROTINA 380R, Hettich, Westphalia, Germany), using the Vivaspin® 20 system (Sartorius®, Stonehouse, UK, 50 kDa molecular weight cut-off) to completely remove the carbon nanoparticles from the food simulants.

#### 2.12.2. Food Simulant Olive Oil

A volume of 5 mL of each CNP suspension was centrifuged at 140,000× *g*, 4 °C for 1 h (Beckman L8-50 M/E Ultracentrifuge; Rotor type 42.1, Brea, CA, USA), and the pellet was collected and resuspended in 20 mL of olive oil. After incubation, all samples were again subjected to ultracentrifugation at 140,000× *g*, 10 °C for 90 min, and the nanoparticle-free supernatant oil was collected.

### 2.13. Quantification of Copper Ions Spontaneously Released into Food Simulants

All samples were thawed at room temperature on the day of the analyses. The amount of copper ions released in each simulant was determined by ICP-OES. Samples containing water and acetic acid as simulants were directly analyzed, while ethanol samples were diluted at a ratio of 1:10 with ultrapure water. In contrast, olive oil samples were prepared following the extraction induced by emulsion breaking (EIEB) method [53,54,55,56]. First, the extracting solution was prepared by dissolving 2.0 g Triton X-100 in 100 mL nitric acid 3% (*v*/*v*). Then, olive oil samples were vigorously shaken, and 7.0 g of each sample was transferred into a 50 mL Falcon tube containing 7 mL extracting solution. A sample of pure olive oil was also prepared with the same procedure and used as a control. The mixtures were vigorously shaken for 30 seconds and placed in a thermostatic bath at 90 °C for 60 min. After emulsion breaking, two phases were observed: an upper phase containing the edible oil and a lower aqueous phase (acid medium) containing the extracted metals. Finally, all samples were centrifuged at 1500× *g*, 25 °C for 10 min (by ROTINA 380R, Hettich, Westphalia, Germany) to better separate the different phases and stored overnight at 4 °C. The following day, the aqueous phase was recovered using a Pasteur pipette and directly analyzed.

### 2.14. Statistical Analysis

Statistical analysis was performed using MedCalc® Statistical Software version 20.009 (MedCalc Software Ltd., Ostend, Belgium; https://www.medcalc.org; 2021, accessed on 15 June 2022). A non-parametric Friedman test was used to evaluate the significance of microbiological data between groups and controls; *p*-value < 0.05 was considered statistically significant. A non-parametric test was carried out between groups with the same antimicrobial solution (CuSO_4_, SCNPs-Cu, LCNPs-Cu) used at different concentrations and with each respective control (optimal growth 100%).

## 3. Results and Discussion

### 3.1. Synthesis and Physical-Chemical Characterization of Carbon Nanoparticles

In this study, a green chemistry fabrication method was used to produce two sets of monodispersed CNPs of different size (SCNP and LCNP). The synthetic protocol was previously optimized, as described in [42], to obtain highly reproducible spherical materials with a structure mainly composed of amorphous carbon. In the present study, these CNPs were tested for the first time for food packaging applications. CNPs were also loaded with copper by incubation with a solution of CuSO_4_. Pristine and loaded nanoparticles were characterized by FE-SEM, TEM, DLS and ELS (see Figure 1 and Appendix A).

The mean Feret’s diameter of SCNP and LCNP, as assessed by FE-SEM, was 88 ± 22 nm and 242 ± 86 nm, respectively. FE-SEM and TEM analysis revealed that all particles exhibited a spherical shape (see Figure 1B,C and Appendix A). DLS analysis indicated a mean hydrodynamic diameter of 181.0 nm for SCNP and 440.8 nm for LCNP (see Figure 1A, Table 2). This is in line with the assessments by FE-SEM, since DLS measures hydrodynamic diameter (dH) in water, yielding results that are larger than the geometrical ones. Furthermore, the technique overestimates larger NPs due to the higher scattering intensity [44]. Since DLS measurements are performed in dispersion, they also make it possible to measure the NP colloidal stability by performing subsequent measurements on the same suspension. In fact, particle agglomeration and sedimentation generate large variation in the size distribution curves and dH mean values. Both samples formed stable colloidal suspensions, as inferred by the small standard deviation (SD) in the DLS patterns in Figure 1A. The low values of the PdI (Table 2) confirmed that the particles in the suspensions had a poor tendency to agglomeration. This was due to the presence of hydrophilic, negatively charged groups at the CNP surfaces formed during the synthesis process [42], providing electrostatic repulsion among particles. This was confirmed by the highly negative ζ-potential values (Table 2) and slightly acidic pH of the suspensions.

The CNP suspensions maintained similar features after being loaded with copper ions (Figure 1A, Table 2). A negligible mean dH and ζ-potential shift was observed after Cu^2+^ loading to SCNP, suggesting that the amount of copper ions bounded to the particles was not enough to saturate the negative charges once the CNPs had been purified by the excess of metal ions. A moderate destabilization of the colloidal suspension was observed for LCNP after loading with Cu^2+^ (Figure 1A), although this became stable over time (data not shown), as was expected, since its ζ-potential value remained highly negative, with a value comparable to that of pristine particles (Table 2).

The amount of loaded copper was evaluated by ICP-OES (see Table 2). SCNP exhibited the greatest capacity to bind Cu ions when expressed per mass and the highest density of ions at the surface. However, despite the different concentrations of CNPs in the two batches, the concentrations of copper in the two suspensions were serendipitously similar in the two batches. Note that the size distributions of LCNP and LCNP-Cu were above the formal upper limit values defining nanomaterials (100 nm) [58].

### 3.2. Antimicrobial Effect of Copper Loaded Nanoparticles

The *L. monocytogenes* species are divided into 13 serovars based on somatic and flagellar antigens. Since 2005, these serovars have been replaced by five genosero groups, as determined by PCR techniques. Although *L. monocytogenes* 1/2 a was historically the most widespread serotype detected in food, serotype 4d is the most prevalent in food processing environments and is most often responsible for food cross-contaminations [2]. Therefore, in this work, we tested the ATCC 19117 strain serotype 4d.

Firstly, *Listeria monocytogenes* growth was studied to identify the maximum growth rate and earlier stationary phase (10^9^ CFU/mL) from which we obtained the concentration (10^5^ CFU/mL) to use in the MIC trials. The results showed the growth kinetics for 72 hours, and its maximum concentration (10^9^ CFU/mL) was determined after incubation for 24 h (Appendix A). The addition of the CNP suspensions to water leads to a dilution of the broth that might induce a reduction of bacterial growth. For this reason, experiments were conducted growing *Listeria monocytogenes* under conditions of reduced nutrients by adding 100, 200 or 300 µL physiological solution (see Appendix A). The CFU/mL value was determined by counting on a LSM plate, and no exponential growth differences were detected between samples and control (1000 mL). In fact, the growth variability was in a range of (4.6 ± 1) × 10^9^ CFU/mL, suggesting that the microorganism was not affected by stressful conditions. Therefore, the corresponding ratio of broth volume was used for the MIC analysis performed with CuSO_4_ · 5 H_2_O (Cu^2+^) and both small and large CNP-Cu.

The optical density method (OD) is commonly used to determine the number of bacteria in a sample quickly and easily, based on turbidity measurements [59]. To evaluate the validity of this method in our research, a spectrometry analysis was carried out to obtain information about the maximum absorption of different concentrations of *L. monocytogenes* (Appendix A). The values were in the range from 427 nm to 350 nm for *L. monocytogenes*, but from 616 to 231 nm for each antimicrobial solution combination (data not shown). Because of the radical changes of the spectra due to the numerous combinations of solutions, suspensions and concentrations and the presence of nano-objects, we could not implement an OD analysis. Therefore, we decided to count the colony forming units (CFU) on a plate.

The effects of soluble copper salt and loaded copper nanoparticles were evaluated on ATCC 19117 strain after incubation for 24 h (Figure 2). Previous findings in the literature on Gram-positive bacteria such as *S. aureus* reported 100% inhibition with 1000 ppm of soluble copper [60]. This evidence was confirmed in our previous assessment on *Listeria monocytogenes* using 1000 ppm, 750 ppm and 500 ppm of soluble copper (see Appendix A).

The doses selected for MIC were chosen to have a similar range of copper concentrations in all experiments (low, medium, high), as indicated in Table 1.

Statistical analyses were executed between groups of the same antimicrobial solution and controls through non-parametric Friedman test with a *p*-value < 0.05. Each experimental value showed a significant difference from the control (data not shown).

Soluble copper induced a clear antimicrobial effect at medium and high concentrations, with the latter corresponding to a growth inhibition of 65%. Unexpectedly, at low concentrations, an increase of bacteria growth was observed. This unusual pro-proliferative effect might be due to the increase of uptake of copper when *Listeria monocytogenes* is exposed to non-toxic copper concentration [23]. As is widely known, copper plays essential roles in biological systems [61], i.e., it acts as a cofactor in enzymes which catalyze a wide variety of redox reactions [62].

A pro-proliferative effect was not observed with either small and large copper loaded CNPs, in which an inhibitory effect of 11% for SCNP-Cu and 30% for LCNP-Cu was observed at similar doses. A clear dose-dependent effect was observed for both loaded CNPs; however, while SCNP-Cu appeared to be less active than the soluble copper, reaching an inhibition of 34% at the highest dose, LCNP-Cu elicited higher activity against the bacteria, with a maximum inhibition of 85%. Such a high rate represents a very important result in the microbiology and foodborne contamination fields. In fact, as the EU Regulation 2073/2005 reports, *Listeria monocytogenes* is safe under the limit of 100 CFU/g in ready-to-eat food if the conditions are unable to support the growth (Category 1.3) or if growth is supported (Category 1.2) but the manufacturer is able to demonstrate that the product will not exceed the limit during its shelf-life [63]. Therefore, a significant inhibition of *L. monocytogenes* is expected for contaminated foods treated with copper loaded CNPs, but challenge tests are necessary to evaluate the effective inhibition rates in complex food matrices.

### 3.3. Mechanism of Action of Loaded CNPs

As shown in the previous paragraph, at equal nominal concentrations of copper, a different antimicrobial potency was observed for loaded CNPs compared to soluble copper, suggesting a role of the nanoparticles in the mechanism of action. Several properties of CNPs may have contributed to the observed effects: a reduction of activity might be due to the lower bioavailability of copper ions bound to the nanoparticle surfaces or a lower reactivity. In fact, Cu is a redox-active metal, and depending on its coordinative and oxidation state, it can generate cytotoxic ROS by Fenton reaction, as follows:Cu^2+^ + H_2_O_2_ → Cu^+^ + 2 H^+^ + O_2_^−•^(1)
Cu^2+^ + O_2_^−•^ → Cu^+^ + O_2_(2)
Cu^+^ + H_2_O_2_ → Cu^2+^ + OH^−^ + HO^•^(3)

These ROS are involved in the mechanism of toxicity of copper [22]. Binding to the nanoparticle surfaces might modify the redox potential and, in turn, decrease the reactivity. On the other hand, CNPs may increase the antimicrobial activity of copper by acting as a carrier of metals inside the cells, as observed with other systems [64], or they may have antibacterial activities themselves [65]. Finally, CNPs may increase the reactivity of copper. In fact, CNPs can reduce Cu^2+^ ions to Cu^1+^ that, in turn, generates hydroxyl radicals by a Fenton-like reaction (reaction (3)), as shown in a previous study [11].

Here, two possible effects of CNPs were explored: the modification of the bioavailability of copper and the modification of its ability to generate ROS. To this purpose, the release of copper ions into bacterial growth medium BHI was evaluated after 24 h of incubation, while the reactivity of the particles, dosed at equal amount of copper, was evaluated by EPR spectroscopy (Figure 3). This technique can detect several ROS if they are trapped with suitable spin trapping agents.

The leaching of copper ions into BHI broth was evaluated after 24 hours at 37 °C, using increasing concentration of CNPs. Our analysis revealed a significant copper release into BHI broth (Figure 3A). As expected, the detected copper concentration increased with increasing the volume of CNPs. However, for both small and large CNPs, a substantial amount of copper remained bound to the CNPs after 24 h.

Figure 3B,C, report the EPR signals recorded on SCNP-Cu and LCNP-Cu suspensions containing the spin trapping agent DMPO in the presence of hydrogen peroxide. The signal of soluble copper was also measured as a control. Intense EPR signals were recorded for SCNP-Cu, LCNP-Cu, and CuSO_4_, while CNPs in their pristine form showed a signal similar to that of the negative control. These results are in agreement with our previous observations [11].

Since the intensity of the EPR signal is proportional to the concentration of free radicals, the data shown in Figure 3 indicate that small and large CNPs have similar reactivities. Moreover, EPR signals of similar intensity were obtained with free copper, suggesting a similar reactivity of bound and free copper ions.

Both an antibacterial effect mediated by a Trojan horse effect and ROS-mediated cytotoxicity are compatible with the reported results. However, they do not explain the observed higher activity of the largest LCNP with respect to both the soluble copper and SCNP. In a previous study [66], we showed that CNPs strongly interact with *E. coli* and *S. aureus* by forming aggregates that might favor the transport of the loaded antimicrobial agents close to the bacterial cells. A similar effect might account for the higher bioactivity of LCNP-Cu; in fact, since the experiments were performed with equal concentrations of copper, the available surface for interactions with bacteria was 10 time higher for LCNP-Cu than with SCNP-Cu. Further studies are necessary to confirm this hypothesis. 

### 3.4. Assessment of the Release of Copper Ions in Food Simulants

In view of the possible application of these materials in the food industry, the release of copper into food needs to be assessed. In fact, the EU Commission Regulation No 10/2011 expressly indicates in Annex II, Table 1, that plastic materials and articles shall not release copper in quantities exceeding 5 mg/kg of food or food simulant (5 ppm). It also provides guidelines on which type of food simulants (Annex III) and experimental conditions (Annex V) should be used in tests to properly mimic food matrices and their storage conditions [36]. Therefore, incubation experiments of SCNP-Cu and SCNP as controls were performed in the dark at 40 °C for 10 days. After incubation, the nanoparticles were removed, and the concentrations of Cu were evaluated by ICP-OES (Figure 4).

Copper was released in water, acetic acid and ethanol, while no release was observed in oil. The highest release was observed in acetic acid (50%). This is in agreement with previous studies reporting the migration of metals in the same simulants [67,68,69,70,71], since a low pH is expected to induce the dissolution of transition metal ions [69]. It can therefore be reasonably assumed that metal release from CNP-Cu in acidic foods will be greater than in foods with a predominantly aqueous or fatty composition. This observation suggests an increased antimicrobial action of these materials in acidic foods, but at the same time, indicates a greater risk of human exposure to metals through food. The equilibrium between these two aspects must be carefully considered.

## 4. Conclusions

The results reported herein demonstrate the antimicrobial activity of copper-loaded nanoparticles against *Listeria monocytogenes*. The particle size was shown to modulate the toxicity, with a higher efficacy being observed with the largest CNPs. As such, large CNPs appear to be the best candidate for the design of antimicrobial food packaging materials.

The loaded CNPs appeared to be able to generate significant amounts of ROS, which may have been involved in their mechanism of action.

When in contact with food simulants, a substantial release of copper ions was observed. This release was found to increase at low pH; as such, we predict a higher release in foods with acidic compositions.

Overall, the present results, while preliminary, support the use of LCNP-Cu as a candidate for the manufacturing of active packaging for the control of *Listeria monocytogenes*.

## Figures and Tables

**Figure 1 foods-11-02941-f001:**
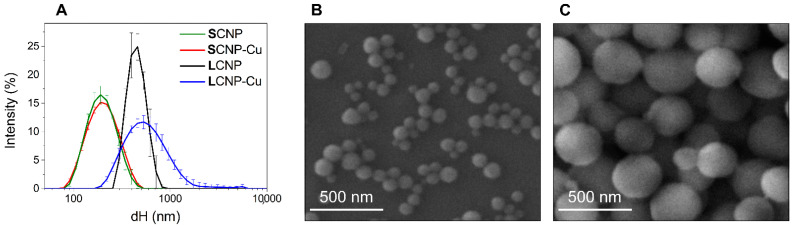
Size distribution of CNPs. (**A**) Hydrodynamic diameter distribution, as evaluated by DLS of pristine and Cu-loaded CNPs suspended in water. Each line is the mean of five measurements of the same suspension. Error bars represent the SD. (**B**,**C**) FE-SEM images of (**B**) SCNP and (**C**) **L**CNP.

**Figure 2 foods-11-02941-f002:**
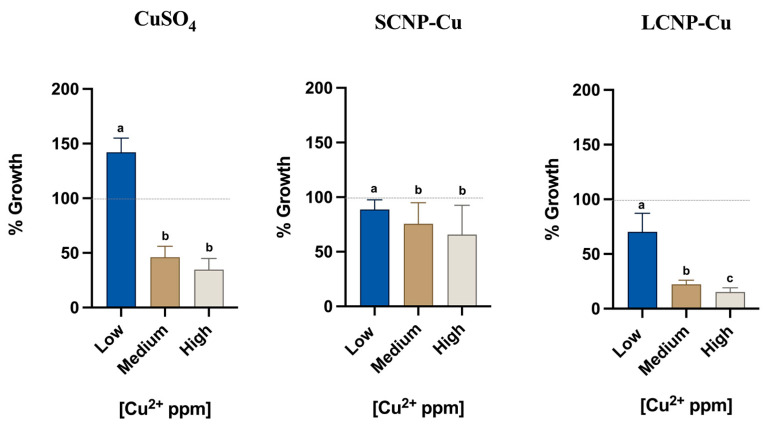
The antimicrobial activity of CuSO_4_, SCNP-Cu e LCNP-Cu (low, medium, high copper concentration) was evaluated against *Listeria monocytogenes*. The optimal growth condition of *Listeria monocytogenes* as a negative control (10^9^ CFU/mL) during the MIC experiments is indicated as 100% on the *y*-axis (dashed gray line). The *x*-axis represents the copper concentration (low, medium and high) for each antimicrobial. Average values are used for the histogram representation, and values with different superscript letters (a, b, c) are significantly different (*p* < 0.05).

**Figure 3 foods-11-02941-f003:**
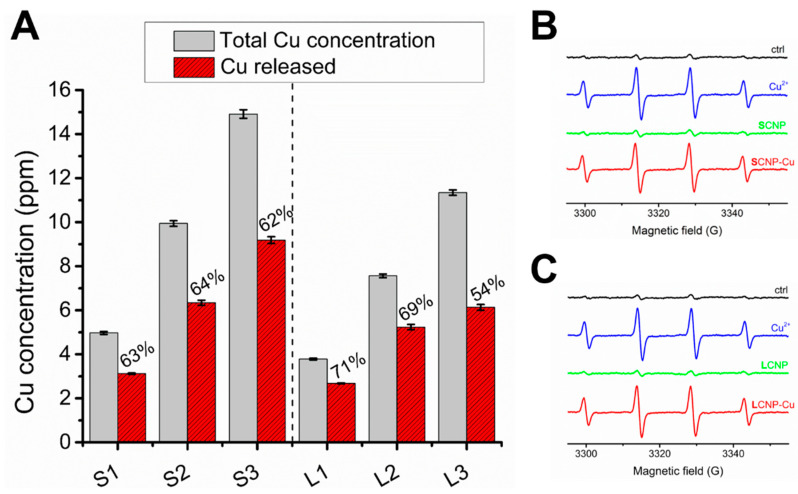
Release of copper in the BHI broth and ROS generation by CNPs. (**A**) Concentration of copper in BHI broth following incubation with three different concentrations of SCNP-Cu (S1–S3) or LCNP-Cu (L1-L3) for 24 h (red columns) detected by ICP-OES in comparison with the theoretical concentration of copper relative to total release (black columns). (**B**,**C**) EPR spectra generated by (**B**) SCNP and SCNP-Cu and (**C**) LCNP and LCNP-Cu after 60 min of incubation in the presence of 88 mM DMPO and 40 mM H_2_O_2_ in 4 mM PBS at pH 7.4.

**Figure 4 foods-11-02941-f004:**
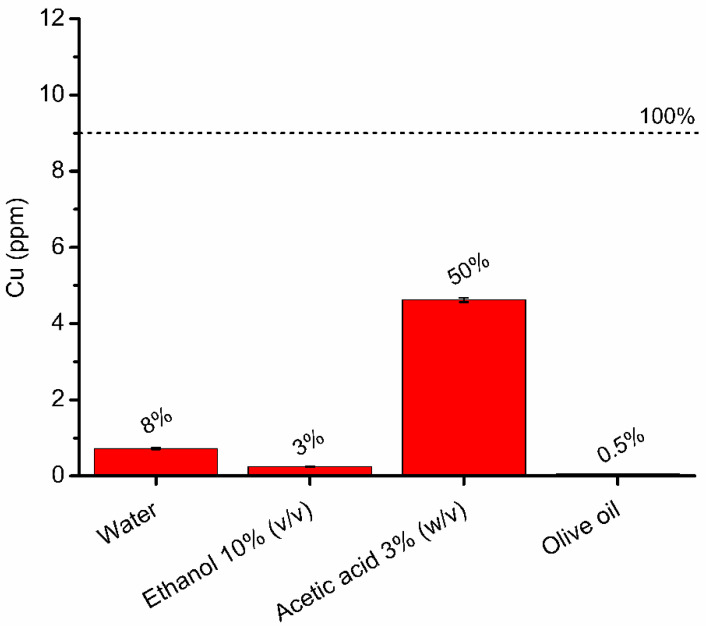
Copper release from SCNP-Cu into each food simulant after 10 days of incubation in the dark at 40 °C. The dotted line represents the total amount of copper loaded on the SCNP-Cu during incubation with food simulants, corresponding to the maximum concentration. Values are expressed as % of CU release.

**Table 1 foods-11-02941-t001:** Antimicrobial ppm concentration of copper used for MIC tests on *L. monocytogenes* (ATCC 19117).

Sample (Cu^2+^ ppm)	CuSO_4_	SCNP-Cu	LCNP-Cu
**Low**	5.0	5.0	3.9
**Medium**	11.1	9.9	7.8
**High**	19.0	14.9	11.6

**Table 2 foods-11-02941-t002:** Properties of the colloidal suspensions of pristine and Cu-loaded CNPs.

	Hydrodynamic Diameter (Z-Average) nm ± SD *	PdI	ζ-Potential (mV)	Amount of Cu Loaded (µg Cu/mg CNP)	Amount of Cu Loaded (Ions Cu/nm^2^ of CNP Surface) **	Concentration Cu in Suspension (ppm)
**SCNP**	181.0 ± 3.1	0.148 ± 0.006	−42.4 ± 0.8	-	-	-
**SCNP-Cu**	185.0 ± 3.6	0.075 ± 0.034	−46.0 ± 0.2	32.48 ± 0.42	7.60 ± 0.10	49.7 ± 0.6
**LCNP**	440.8 ± 13.6	0.036 ± 0.039	−37.4 ± 0.7	-	-	-
**LCNP-Cu**	511.0 ± 8.1	0.086 ± 0.042	−37.0 ± 1.4	1.13 ± 0.01	0.74 ± 0.01	37.9 ± 0.4

* Each value is the mean of five measurements on the same suspension ± standard deviation (SD). ** Calculated by considering a spherical shape and a density of 1.68 g/cm^3^ (as previously reported by Soddu et al. [57]) and the mean of geometrical diameter evaluated by FE-SEM.

## Data Availability

Data are available upon request.

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
