# Peer review of "Inhibitory Effect against Listeria monocytogenes of Carbon Nanoparticles Loaded with Copper as Precursors of Food Active Packaging"

_foods, 2022, doi:10.3390/foods11192941_

Round 1

Reviewer 1 Report

These are my suggestions for the paper entitled "Inhibitory effect against Listeria monocytogenes of carbon nanoparticles loaded with copper as precursors of food active packaging":

1) better explanation of cooper as antimicrobial need to be part of introduction part; add actual references and explain mechanism of the inhibitory effect

2) the novelty is missing in the introduction part as reason why this study was done; please, add.

3) Add manufacturers, cities and states in Material and methods for all chemicals and devices used

4) Add the full name of the ATCC collection as well as necessary details from the collection about the used bacterium

5) add an explanation of how did you refresh a bacterium culture, prepare a stock solution, determine the concentration, etc.

6) better quality Figures are required. 

7) can you add some statistical analysis in Table 2? 

Author Response

All the comments are considered as useful to improve the authors knowledges. The answer of each comment and the appropriate corrective action of the authors are reported in the attachment file.

Reviewer 2 Report

L176 Avoid starting the sentence with number. Should be Five

Table 1 Any replication? If so, please add SD

L372-373 Why can SD explain stability of colloid?

Table 2 Add statistical analysis

Fig. 2 How the Y-axis was calculated?

Fig. 3 Add error bars

L521 Add more discussion e.g., Metal nanoparticles commonly had antimicrobial efficiency and photocatalytic activity including capability to eliminate oxygen and ethylene gas due to oxidative degradation (doi.org/10.3390/polym13234192).

L544 Previous investigation also indicated the highest release of Zn2+ ion from biodegradable films in 3% acetic acid solutions as with aqueous simulants (doi.org/10.1016/j.fpsl.2022.100901).

L556 Should be revised -> Conclusions should mainly focus the present results.

Author Response

(The authors gave the same response as above.)

Round 2

Reviewer 2 Report

The manuscript has been improved.